# Polysaccharides’ Structures and Functions in Biofilm Architecture of Antimicrobial-Resistant (AMR) Pathogens

**DOI:** 10.3390/ijms24044030

**Published:** 2023-02-17

**Authors:** Evita Balducci, Francesco Papi, Daniela Eloisa Capialbi, Linda Del Bino

**Affiliations:** 1GSK, 53100 Siena, Italy; 2Department of Biotechnology, Chemistry and Pharmacy, University of Siena, 53100 Siena, Italy

**Keywords:** exopolysaccharides, biofilm, AMR

## Abstract

Bacteria and fungi have developed resistance to the existing therapies such as antibiotics and antifungal drugs, and multiple mechanisms are mediating this resistance. Among these, the formation of an extracellular matrix embedding different bacterial cells, called biofilm, is an effective strategy through which bacterial and fungal cells are establishing a relationship in a unique environment. The biofilm provides them the possibility to transfer genes conferring resistance, to prevent them from desiccation and to impede the penetration of antibiotics or antifungal drugs. Biofilms are formed of several constituents including extracellular DNA, proteins and polysaccharides. Depending on the bacteria, different polysaccharides form the biofilm matrix in different microorganisms, some of them involved in the first stage of cells’ attachment to surfaces and to each other, and some responsible for giving the biofilm structure resistance and stability. In this review, we describe the structure and the role of different polysaccharides in bacterial and fungal biofilms, we revise the analytical methods to characterize them quantitatively and qualitatively and finally we provide an overview of potential new antimicrobial therapies able to inhibit biofilm formation by targeting exopolysaccharides.

## 1. Introduction

Bacterial cells exist in two different states: a planktonic state (free-floating) and a sessile state, where the bacteria adhere to a surface. As a consequence of the bacterial cells’ adherence to a surface, genes responsible for the biosynthesis and maturation of the extracellular substance are altered. This process determines the formation of a protective barrier embedding bacterial cells, which is called biofilm. The traditional definition of biofilm refers to a structured polymeric matrix, adherent to a surface and enclosing bacterial cells [1]. Biofilms are an important surgical issue since they can form on a chronic wound or implantable medical devices such as prosthesis, pace-makers, hearth valve, catheters, etc., making them therefore one of the most common risk factors for infections. Both Gram-positive and Gram-negative bacteria can form biofilms, but the ESKAPE pathogens (*Enterococcus faecium*, *Staphylococcus aureus*, *Klebsiella Pneumoniae*, *Pseudomonas aeruginosa*, *Acinetobacter baumannii* and *Enterobacter* species) are the leading cause of nosocomial infections throughout the world [2,3,4].

The formation of a mature biofilm is a process that traditionally has been described as composed of five phases, as shown in Figure 1: (1) reversible attachment phase of bacteria onto a surface through non-specific weak hydrophobic interactions; (2) irreversible attachment phase, involving adhesion of the bacteria through adhesins such as fimbriae and lipopolysaccharides; (3) production of extracellular polymeric substances (EPS) by the resident bacterial cells; (4) biofilm maturation phase, when bacterial cells synthesize and release signaling molecules to send messages to each other, conducting the formation of microcolonies and biofilm maturation; (5) dispersal/detachment phase, where the bacterial cells detach from biofilms and come back to the planktonic state [5]. Recently, a new, simpler biofilm model, going beyond the classical five-step model, was proposed [6].

Bacterial EPS mostly consist of exopolysaccharides, proteins, lipids and extracellular DNA (eDNA) [7]. The formation of EPS requires a high investment from cell resources and energy, but it provides bacterial cells an environment with unique functionalities: EPS favor horizontal gene transfer and inter-cellular interactions, improve the resource capture and the surface adhesion, offer protection against external agents and inhibit bacterial dehydration [8]. Biofilm formation is also associated with higher antibiotic resistance due to several factors, such as failed antibiotic penetration in the biofilm matrix, the presence of the bacterial efflux pump to push antibiotics outside the biofilm and interactions between multiple species [9,10,11,12,13,14,15,16]. Polysaccharides are a major fraction of the EPS matrix both in Gram-positive and Gram-negative bacteria [17]. Several exopolysaccharides are homopolysaccharides (glucans, fructans, cellulose); however, most of them are heteropolysaccharides that consist of a mixture of neutral and charged sugar residues. Many known exopolysaccharides, including Alginate, xanthan and colanic acid, are polyanionic. Polycationic exopolysaccharides also exist, such as PNAG (Poly-β-(1→6)-N-acetylglucosamine) and Pel.

Polysaccharides in the EPS are involved in several functions in the formation and maintenance of biofilm: they allow the colonization of biotic and abiotic surfaces by planktonic cells and enable the aggregation of bacteria; they form a hydrated polymer network that mediates the mechanical stability of biofilm and prevents bacterial desiccation; they confer resistance to specific and non-specific host defense during infection; and they are a source of nutrients [18].

Interfering with biofilm formation is an attractive strategy to develop new therapeutics. Understanding the composition of biofilms and the function of each component is pivotal to designe new preventive therapeutic tools against infections caused by microbial biofilms. Herein, we describe the exopolysaccharides which are most commonly found in the biofilm matrix of pathogens causing infections in humans, their functions in forming and maintaining the EPS matrix, advancements in the characterization methods of the polysaccharides in the biofilm and their potential role as therapeutic targets.

## 2. Biofilm Polysaccharides

### 2.1. PNAG (PIA)

PNAG is a high molecular weight polysaccharide consisting of β(1,6)-linked *N*-Acetylglucosamines; in its mature form, about 10 to 20% of the amino groups are not acetylated [19]. It is also referred to as PIA (polysaccharide inter-cellular adhesin) due to its function [20]. 

It was first discovered as a major component of the biofilm formed by *S. aureus* and some other staphylococci [21,22], but later this exopolysaccharide was also found to be produced by pathogens of various natures [23,24,25,26,27]. The PNAG biosynthesis pathway has indeed been characterized in different species: in *S. aureus* and *S. epidermidis*, PNAG is synthesized by proteins encoded by the *icaADBC* genes of the inter-cellular adhesin locus (*ica*), while *E.coli* has a homologous genetic locus, named *pgaABCD* [23,28]. 

Homologues of the Gram-negative *pga* locus have been described in several other important pathogens, even if PNAG production by these species was not demonstrated [23]. On the other hand, the presence of PNAG was immunochemically detected on several prokaryotic and eukaryotic cells, including bacteria, fungi and protozoa, for which the genetic locus indicating PNAG biosynthesis has not been found yet [29]. In this way, PNAG has been identified as a conserved surface polysaccharide produced by major bacterial, fungal and protozoal pathogens that can all be targeted for vaccination using this single antigen. In *S. aureus*, *S. epidermidis*, *E. coli*, *Yersinia pestis*, *Actinobacillus pleuropneumoniae* and *Acinetobacter baumannii*, the genes for biosynthetic proteins were identified. Many microbial pathogens lacking an identifiable *ica* or *pga* locus produced PNAG too: in these cases, the PNAG production was detected by an immunomicroscopy method using the anti-PNAG mAb F598 [29]. Overall, the synthesis of PNAG by both Gram-positive and Gram-negative pathogens suggests an important role for this molecule in microbial biology.

In *S. aureus* and *S. epidermidis*, PNAG is synthesized by the proteins IcaA, IcaD, IcaB and IcaC that are encoded within the inter-cellular adhesin (*ica*) operon. IcaA is a trans-membrane *N*-glucosyltransferase that initiates the polymerization of short PNAG saccharides using UPD-*N*-acetyl glucosamine as a substrate, and then the small inner membrane protein IcaD increases the biosynthetic efficiency of IcaA. IcaC is a trans-membrane as well and seems to be involved in the linkage of short oligosaccharides to synthesize longer polysaccharides. The polymer is then transported through the periplasmic space to the *N*-deacetylase IcaB that removes 5–40% of the *N*-linked acetates, and then the final polymer is anchored to the cell surface [30,31]. This IcaB-induced partial deacetylation of PNAG is needed for its association with the negatively charged bacterial cell surface, which is necessary for biofilm formation and for the evasion of phagocytosis [22]. 

PNAG is a critical virulence determinant in infections caused by *S. aureus*, *S. epidermidis* and *E. coli* as it is involved not only in biofilm formation but also in adherence to polymeric substrates and biomaterials, bacterial inter-cellular adhesion [32,33] and protection against antibody-independent opsonic killing [22,34]. PNAG has a fundamental role in the main steps of the biofilm formation process: the attachment of microbial cells to a surface, the proliferation and maturation of the biofilm and the final detachment of microbial cells or clusters [31,34]. 

The impact of PNAG on infection was studied mostly in *S. epidermidis* using isogenic deletion mutants. Firstly, Rupp et al. demonstrated a relevant impact of the *ica* genes on catheter-associated infection in mice and rats [35]; later, PNAG’s role was also investigated in orthopedic infections taking into account several biomaterials [33]. Moreover, Vuong et al. showed that the cationic character of PNAG, conferred by the partial deacetylation, is essential for many of the biofilm formation steps; using an isogenic IcaB mutant strain, they were able to demonstrate that the loss of PNAG’s cationic character made devoid the ability to attach to the cell surface, and that it is essential for biofilm formation, colonization and resistance to phagocytosis [22]. 

In the phase of accumulation and biofilm formation, PNAG mediates the attachments of staphylococcal bacteria in a fibrous net and builds up biofilm mass. The formed biofilm has increased resistance to mechanical force, confirming the crucial role of PNAG for biofilm formation under high-shear flow conditions such as those found inside catheters [15,35,36]. 

In the final stage of biofilm formation, staphylococcal cells are detached from the biofilm, via mechanic force, by enzymatic digestion via proteases or nucleases or via detergents. Biofilm detachment with consequent bacterial dissemination is caused by staphylococcal secretion of quorum-sensing molecules, such as phenol-soluble modulins [37]. Conversely, PNAG-degrading enzymes have not been found in staphylococci: the only known enzyme to degrade PNAG is dispersin B, which has never been detected in staphylococci [38].

### 2.2. Alginate, Psl, Pel

*P. aeruginosa* is genetically capable of producing three distinct exopolysaccharides, Alginate, Psl and Pel [39], that are greatly involved in surface attachment, formation and the stability of biofilm architecture [40]. Each of these polysaccharides is associated with some stages of *P. aeruginosa* biofilm development, and also with different phases of the infectious process [41]: Psl and Pel are involved in the initial stages of biofilm development (acute stages of infection), while Alginate is the “stress” response polysaccharide associated with chronic infections. Aggregates of non-mucoid clinical isolates in cystic fibrosis (CF) airways have recently been demonstrated to express both Pel and Psl [42].

Psl was identified by not producing Alginate strains but still being able to form biofilm structures on solid glass surfaces [43,44,45]. Psl consists of a repeating neutral pentasaccharide containing D-glucose, D-mannose and L-rhamnose moieties [46]. This exopolysaccharide is crucial for the adhesion of sessile cells to surfaces and cell-to-cell interactions during biofilm initiation of both non-mucoid and mucoid strains [47,48]. Psl forms fibrous structures that are organized in a helical pattern around *P. aeruginosa* cells, creating a mesh-like structure to which neighboring cells can bind. Psl tracks have been described behind migrating cells and are supposed to provide a guided trail for bacteria that explore the area, contributing to the aggregation of microcolonies [49]. In a mature biofilm, Psl is situated in peripheries of the mushroom-like structure where it contributes to keeping structural stability [50]. Increased Psl expression contributes to cell aggregates’ formation in a liquid culture as has been observed in CF patients’ sputum [51]. Psl promotes the production of c-di-GMP (bis-(30-50)-cyclic dimeric guanosine monophosphate) that is responsible for thick and robust biofilms [52]. Given its implications in the attachment, maintenance and maturation of the biofilm structure, the downregulation of Psl results in sparse and flimsy biofilms [53,54]. In addition, Psl masks biofilm bacteria from antimicrobials [55] and neutrophil phagocytosis [56], constituting a first-line defense to reach persistent infection.

Pel is a partially de-*N*-acetylated linear polymer of α-1,4-*N*-acetylgalactosamine comprised predominantly of dimeric repeats of galactosamine and *N*-acetylgalactosamine [57]. As with Psl, Pel is a key component of biofilm in non-mucoid strains and is involved in the surface attachment phase, as well as the maintenance of biofilm integrity [58,59]. Pel forms the pellicle biofilm which is present at the air–liquid interface of a static broth culture [60]. A pellicle is a “floating biofilm” which provides the colony access to high concentrations of oxygen and nutrients and can be seen on the surface of standing cultures. Pel can serve two functions in biofilms: it is able to play both a structural and a protective role [61]. It is necessary for maintaining cell-to-cell interactions in the biofilm of PA14 naturally Psl-deficient strain, acting as a primary structural scaffold for the bacterial cells. The deletion of pelB in that strain results in a pronounced biofilm deficiency with a strain-specific effect. Indeed, the loss of Pel production in strain PAO1, which produces Psl-dominant biofilms, does not result in attachment or biofilm development modifications [62]. Furthermore, Pel plays a second role by promoting the resistance to aminoglycoside and colistin antibiotics for biofilm-embedded bacteria [62], and Pel-containing biofilms are less susceptible to killing mediated by neutrophils derived from human HL-60 cell lines [63].

The synthesis of Psl and Pel is strain-specific and switches in response to environmental conditions. It has been hypothesized that Pel and Psl are structurally redundant, resulting in *P. aeruginosa* strains often expressing one or the other [61]. The absence of Psl enhances the production of Pel, and the lack of Pel enhances the production of Alginate [40]. In vitro studies showed that Pel and Psl carry out many functions including surface attachment, structural integrity of the biofilm and antimicrobial and immune tolerance [64]; however, it is still unclear how different exopolysaccharides impact *P. aeruginosa* infection or the host response. Recently, the impact of the absence of the two exopolysaccharides Psl and Pel was examined in a mouse wound infection model. The production of Pel and Psl seems to not impact *P. aeruginosa* pathogenesis in mouse wound infections, but they may play a significative role for bacterial persistence in vivo [65]. 

Alginate was the first discovered *P. aeruginosa* exopolysaccharide, associated with the mucoid strains isolated from CF patients [66], and it is the best characterized exopolysaccharide produced by *P. aeruginosa*. Alginate is especially linked to chronic infections and used for biofilm formation by mucoid *P. aeruginosa* strains due to a mutation in *mucA22* allele [67]. The mucoid phenotypes are found mainly in CF isolates, indicating the conversion from acute to chronic infection [68,69]. Alginate from *P. aeruginosa* is a high-molecular-mass unbranched co-polymer consisting of the (1-4)-linked uronic acid residues of β-D-mannuronate and its C-5 epimer α-L-guluronate [70]; these components are organized in homopolymeric blocks of poly β-D-mannuronate and heteropolymeric sequences with random distribution of mannuronate and guluronate residues. *P. aeruginosa* Alginates are partially *O*-acetylated at the O-2 and/or O-3 positions of the mannuronate residues [71]. The ratio of mannuronic to guluronic acid varies from strain to strain, on the order of 10:1 to 1:1 [71,72]. The ratios between mannuronic acid and guluronic acid influence the viscoelastic properties of biofilms which prevent cough clearance in CF patients’ lungs infected with *P. aeruginosa* [73,74]. *O*-acetyl substituents are likely one of the important components of Alginate protecting mucoid *P. aeruginosa* from the host defenses [75]. Although Alginate production has been shown to be not essential for biofilm formation and to not play a relevant role outside of the CF lung [43], studies have revealed that it does play a role in the maturation of biofilms and in the formation of thick and highly structured biofilms with differentiated microcolonies [76,77]. *O*-acetyl groups seem to be needed for cell aggregation functions during early stages of surface colonization, and also for maintaining the shape of macrocolonies in mature biofilms. *O*-acetylation appears to confer glue-like properties on Alginate molecules, thus mediating strong interactions between neighboring biofilm cells, which cannot be overcome by motility [78]. When acetyl groups are not present, Alginate does not have these cohesive properties, and acetylation-defective bacteria are unable to create solid contacts and could freely move away. De-*O*-acetylation can lead to a conformational change of Alginate [79] and/or affect the binding forces between Alginate molecules [80]. There is a distinct correlation between the appearance of mucoid *P. aeruginosa* and a worsening clinical prognosis for CF patients [81]. Alginate overproduction by mucoid *P. aeruginosa* can modify biofilm architecture, leading to more antimicrobial-resistant biofilms, thus confirming that the conversion to mucoidal plays a protective role for the bacterial population [82]. Alginate contributes to establish biofilms upon adhesion of cells to solid surfaces [83,84]. It has also a protective role, since it confers protection against antibiotics [85] and oxidative agents such as sodium hypochlorite [86]. Moreover, Alginate has a key role in the resistance of *P. aeruginosa* to the oxidative stress generated by lethal UVA doses, both in planktonic cells and in static biofilms [87]. Inflammatory cells are recruited in infected CF, where they secrete reactive oxygen species causing extensive tissue damage. Alginate protects *P. aeruginosa* from the consequences of this inflammation, since it scavenges free radicals released by activated macrophages in vitro and seems also to protect against phagocytic clearance [88,89].

### 2.3. Galactosaminogalactans and Galactomannans

*Aspergillus fumigatus* is a mold existing in two different morphotypes: as conidium, that first enters the host system and is responsible for establishing the infection; and hyphae, whose cell wall is structurally different. The outer layer of the hyphae contains galactosaminogalactan: a heterogeneous linear polymer composed of galactopyranose and N-acetylgalactosamine linked with α(1,4)-linkages [90].

*A. fumigatus* establishes and maintains pulmonary infection through the production of biofilms, which are formed especially during invasive infections in immunocompromised individuals and airway infections in patients with chronic lung disease. These biofilms are characterized by mycelia embedded in an extracellular matrix called mycetoma. *A. fumigatus* extracellular matrix (ECM) composition and characteristics have been investigated in vivo and in vitro. Immunocytochemistry studies on *Aspergillus* biofilms established in human lung aspergilloma and mouse lung with invasive aspergillosis (IA) indicated that two polysaccharides, galactomannans and galactosaminogalactans (GAGs), were present in the ECM of both lungs. Polysaccharide α-1,3-glucans were detected in the ECM of aspergilloma biofilms, while in IA, α-1,3-glucans were only found in the inner layer of the hyphal cell wall. Additionally, the polysaccharides galactomannans, GAGs and α-1,3-glucans were detected in an in vitro biofilm model using polystyrene microtiter plates filled with a liquid medium cultured also in the presence of human bronchial epithelial cells obtained from healthy or CF patients and grown in a static condition [91,92,93,94,95]. 

The exopolysaccharide GAG has a relevant role in both biofilm formation and modulation of the immune response during invasive infection. It is an adhesin molecule of *A. fumigatus* and mediates adherence to plastic, fibronectin and epithelial cells. GAG is a heteropolysaccharide composed of α-1,4-linked galactose, *N*-acetylgalactosamine and galactosamine, secreted by actively growing hyphae. The biosynthetic pathway associated with GAG production comprises a cluster of five co-regulated genes coding for enzymes involved in polysaccharides’ synthesis. One of these enzymes, the GalNAc deacetylase Agd3, has been characterized, and its presence demonstrates that the deacetylation of GAG is required for it to adhere to substrates and for full virulence. These studies also suggested that GAG-mediated adherence is a consequence of charge–charge interactions between the polycation polysaccharide and negatively charged surfaces [96]. 

GAG is produced in both chronic and invasive infection. GAG-deficient strains do not form biofilms and are less virulent in the mouse models of invasive aspergillosis. It has also been demonstrated that GAG plays a role in the virulence of the fungus in vivo, since it favors fungal growth and it has anti-inflammatory properties: it blocks IL-1 signaling and induces neutrophil apoptosis via an NK-cell-dependent mechanism [97,98]. 

GAG-specific antibodies are found in the majority of the human population, and infection with *A. fumigatus* does not determine a change in anti-GAG serum titers. Since the role of GAGs as targets of antifungal therapies is still debated, structural analyses aiming to elucidate the structure–immunogenicity relationship of GAG structures are needed. To this aim, GAGs’ synthetic oligomers incorporating the possible natural structural variations have been prepared and used for NMR structural studies corroborated by molecular dynamics simulations. These studies revealed that the oligomers adopt an elongated structure, stabilized by inter-residue H-bonds; the C-2 substituents of the oligomers are almost perpendicular to the oligosaccharide main chain axis, presented to the bulk solvents and therefore available for interactions with antibodies [99]. 

The other two exopolysaccharides composing *Aspergillus* biofilm are galactomannan, a β-(1,5)-galacto-α-(1,6)-mannan, which is also the most useful diagnostic marker in patients with IA, and α-1,3 glucans. Galactomannans contain a mannan-binding domain that is bound by DC-SIGN present on the surface of both dendritic cells and macrophages. The galactomannan-mediated increased secretion of IL-10 suggests that the resulting silencing inflammation may favor the initial colonization of *A. fumigatus* in the lung. *A. fumigatus* α-1,3 glucan is formed by linear chains of α-1,3 glucan with intra-chain α(1,4)-linked glucose units for every hundred α-1,3-linked glucose units. It functions as an adhesin and mediates agglutination between swollen conidia or hyphae through interactions between α-1,3 glucan chains. The role of α-1,3 glucan during infection depends on whether this molecule is tested separately or together with other polysaccharides of the cell wall. When tested as a vaccine in mice models, α-1,3 glucan is protective because it induces an anti-*A. fumigatus* Th1-mediated response. However, a mutant deficient of α-1,3 glucan was less virulent than the parental strain in an experimental mouse model of aspergillosis, indicating its role in the proper assembly of the cell wall [100]. 

### 2.4. Other Polysaccharides

The role of extracellular polysaccharides in biofilm architecture has been described for several bacteria. Various exopolysaccharides have been isolated, characterized, and their structure, as well as their role in biofilm formation, elucidated. 

*Vibrio cholerae* is a Gram-negative motile bacterium responsible for water-borne diarrheal disease, which can switch between motile and biofilm lifestyles. It has been demonstrated the *V. cholerae* forms biofilms during infections, and that it is a major virulence factor. A major event in the transition from the planktonic to biofilm lifestyle is the downregulation of motility gene expression and induction of genes required for the biosynthesis of the biofilm extracellular matrix [101].

One of the main components of *V. cholerae* biofilm is the Vibrio polysaccharide (VPS), which was characterized by Yildiz et al. by NMR and GC-MS and constitutes about 50% of the extracellular matrix [102]. The VPS forms the biofilm architecture of *V. cholerae* together with the well-characterized matrix proteins RbmA, RbmC and Bap1. Recent studies characterized the interactions of the VPS with RbmA, highlighting that RbmA binds to the VPS and that RbmA has a bistable switch that influences the formation of higher order VPS-RbmA oligomers, the process of cell–cell adhesion and finally biofilm architecture [103]. Recently, interactions of eDNA with the VPS in the extracellular matrix, which seem to determine the stability and strength of the biofilm, have been studied [104].

*Streptococcus mutans* is one of the main etiological factors of dental caries, and it is able to colonize oral cavities and form biofilms, that are composed, similarly to biofilms from other *Streptococcus* spp. causing dental biofilms, mainly by saliva proteins but with the significant presence of glucans (10–20% of the solid matrix) and fructans (1–2%) [105]. Fructans and glucans mediate the adhesion of bacterial cells to human saliva. In the absence of fructans, bacteria can adhere through alternative adhesion mechanisms, but when fructans are present, they may facilitate bacterial adhesion to form a dental biofilm [106].

*E. coli* is able to colonize medical devices, such as catheters and prosthesis; among the polysaccharides involved in *E. coli* biofilm, other than the beforehand-mentioned PNAG, there are colanic acid and cellulose [107]. Colanic acid is a negatively charged polymer of glucose, galactose, fucose and glucuronic acid that encapsulates the bacterial cell surface, and it is expressed by *E. coli*, *Salmonella enterica* and other gammaproteobacteria [108]. Hanna et al. investigated the role of colanic acid, comparing colanic acid-negative mutant strains. Colanic acid-deficient mutants lost their ability to attach to abiotic surfaces compared to the isogenic wild-type strain [109,110]. 

Another type of exopolysaccharide with a demonstrated role in biofilm is cellulose, which, chemically, is a linear polymer composed of β-1,4-linked glucose residues. Within the polymer, each glucose unit forms two hydrogen bonds with each of its neighbors and these interactions confer incredible stability contributing to the integrity of plant walls [111]. Cellulose is the most abundant polymer on earth, and it is mainly produced by plants; however, some microorganisms are also main producers of the amorphous aggregates of cellulose, as the secreted polysaccharide part of the bacterium extracellular matrix. *E. coli* and *Salmonella* spp. are among the most studied pathogenic bacteria producing cellulose: in both, the genetic and protein machinery for cellulose production include the cellulose synthase genes *bcsA* and *bcsB*, which are regulated by the bacterial second messenger cyclic diguanylate monophosphate (c-di-GMP) [111]. The initial discovery of cellulose production resulted from the investigation of the *E. coli* rdar (red, dry and rough) phenotype, which highlighted how cellulose is co-expressed with amyloid curli fibers forming a nanocomposite that encapsulates individual cells in supramolecular structures. Both curli and cellulose are important for cell–cell interactions and to adhere to abiotic surfaces. They also contribute to the architecture of the resilient biofilm that can influence fitness and virulence [112]. The production of both cellulose and curli in *Salmonella* species is key for the survival and persistence of *Salmonella* on surface environments [113]. Recently, it has been discovered that *E. coli* and *Salmonella* species produce a chemically modified cellulose with a phosphoethanolamino group on the glucose 6-OH. This new zwitterionic polymer has been described by use of solid- and solution-phase NMR, and the enzyme responsible for the installation of the phosphoethanolamino groups, BcsG, has been characterized. The modification with phosphoethanolamino groups seems to have multiple functions in the extracellular matrix: it is required for the formation of long cellulose fibrils and a tight nanocomposite with curli fibers, it may confer resistance against attacks by cellulase-producing microorganisms and it can prevent curli from hyper stimulating immune responses [114].

In *S. aureus* and other Gram-positive bacteria, beyond their structural function, teichoic acids (TAs) play an important role in biofilm formation, significantly enhancing the adhesion of the bacteria to surfaces and biomaterials. TAs comprise wall teichoic acids (WTAs), covalently linked to the peptidoglycan, and membrane-anchored lipoteichoic acids (LTAs) [115]. Because of their structure, they are usually referred to as anionic glycopolymers: their structures vary widely between bacterial species, but it is considered that, except for some unusual strains, WTAs consist of ribitol phosphate repeating units, while LTAs are poly-glycerol phosphate chains. Both have a highly variable content of D-alanine ester and *N*-acetylglucosamine substitutions that reduce the net negative charge of TAs [116,117]. Vergara-Irigaray et al. studied whether PNAG attachment to a bacterial surface is due to ionic interactions with the negative charge of WTAs. However, they concluded that the absence of WTAs had little effect on PNAG production or anchoring to the cell surface, but it did impact the biofilm’s forming capacity and cell aggregative behavior in *S. aureus* [118]. Depletion of WTAs markedly decreased the in vitro capacity of *S. aureus* to form biofilm and favored cell-to-cell interactions making cells aggregate and sediment. This also confirmed the role of TAs in cell aggregation, already described for several bacterial isolates and related to the content of alanine esters and bound divalent metal cations, especially Ca^2+^ [115]. The deficiency in primary attachment to abiotic surfaces could be a consequence of this aggregation, since the mutant cells tend to interact between themselves rather than with artificial surfaces [118]. Additionally, the importance of D-alanine substituents of TAs in bacterial pathogenesis was investigated. Peschel et al. described how a mutant of *S. aureus* lacking D-alanine esters in its TA showed a decreased ability to adhere to plastic and glass surfaces in comparison to the wild-type strain. This is because the stronger net negative charge of the mutant likely brings a significant increase in the repulsive forces, thereby disabling any adherence of the bacteria to polystyrene or glass [119]. 

It is also worth mentioning that extracellular polysaccharides are relevant for pathogen-infecting plants, such as *Xanthomonas campestris pathovar campestris*, the causal agent of black rot disease of cruciferous plants. Its virulence depends on a number of factors, including the biosynthesis of extracellular polysaccharides, named xanthans [120], which suppress plant defenses and play an essential role to initiate the process of attachment needed for biofilm formation. Bianco et al. analyzed several mutants of *X. campestris pathovar campestris* producing structurally different xanthans to understand the structural basis of their virulence, and they found that pyruvilation is essential for *X. campestris* virulence [121]. 

Finally, β-glucans have also been demonstrated to form biofilms in fungi, such as *Candida* spp. and *A. fumigatus*, as a resistance mechanism to antifungal drugs. The enzymes involved in glucans biosynthesis in *Candida* have been identified and characterized, and it has been demonstrated that mutants lacking these genes exhibit a higher susceptibility to antifungals during biofilm growth [122].

In Table 1, polysaccharides forming bacterial and fungal biofilms are described. 

## 3. Characterization of Exopolysaccharides

A plethora of methods to characterize polysaccharides in biofilms have been described. Due to the heterogeneity, different complexity and applicability of various (bio-)chemical techniques employed, a direct comparison is challenging. A combination of multidisciplinary techniques is indeed required in order to elucidate biofilm exopolysaccharides’ composition.

### 3.1. Protein-Based Methods

Classical, non-destructive, microscopic protein-based approaches for exopolysaccharides’ imaging are represented by epifluorescence microscopy and the more versatile confocal laser scanning microscopy (CLSM). Confocal laser scanning microscope-fluorescent lectin-binding analysis (CLSM-FLBA) is an application of CLSM for obtaining high-resolution optical images with deep selectivity by using lectins, a class of carbohydrate-binding proteins which bind to particular sugars through conserved carbohydrate-recognition domains (CRDs) [123]. As a result of their specificity and the need of few sample manipulations, fluorescent-labelled lectins have been suggested as effective CLSM in situ probes and can be used not only for visualization but also for the semiquantitative analysis. The usefulness of various lectins has been widely demonstrated, starting from the most frequently used concanavalin A (ConA) and wheat germ agglutinin (WGA) [124]. Lectin selection is the key point of exopolysaccharides’ detection; however, due to their narrow specificity, the choice could be challenging. Moreover, in order to consider the whole matrix, a combination of multiple fluorescent lectins is needed (Table 2), and this is sometimes complex due to possible overlapping of the emission and excitation wavelengths of fluorophores [125]. While these methods are very reliable for the study of thin or young biofilms, a major limitation of CLSM is represented by the attenuation of the light intensity by the biofilm matrix thickness and compactness leading to an underestimation of the exopolysaccharide contribution due to incomplete lectin staining [126]. A similar approach based on lectin specificity is represented by the application of peroxidase- and alkaline phosphatase-labelled lectins in combination with a enzyme-linked lectinsorbent assay (ELLA) [127,128,129]. Optical detection was also reported using specific enzymes. An active site mutant of the PNAG-specific glycosyl hydrolase dispersin B was fused to an EGFP-DspBE_184Q_ protein, to generate a macromolecular fluorescent-binding probe that bound PNAG but did not hydrolyze it. The DspB-based PNAG probe possessed low affinity for shorter oligomeric forms of PNAG but was highly competent in the binding of the cell surface, making it possible to study PNAG in the context of maturing biofilms [130].

### 3.2. Chromatographic Techniques

Analytical approaches require the physical, chemical and/or enzymatical breakup of the biofilm matrix and extraction of polysaccharide constituents, which must be separated from DNA, proteins and lipids (few exceptions for colorimetric assay vide infra). The isolation and analysis of exopolysaccharides are time-consuming and challenging tasks, as they can affect the composition and change their properties [131]. Additionally, the analysis outcome depends strongly on the extraction protocol and analytical method used, and there is no universal approach which is recommended. In order to analyze the structure and composition of purified exopolysaccharides, chromatographic, spectroscopic and colorimetric methods are frequently used. 

The molecular mass determination of purified exopolysaccharides can be performed by size exclusion chromatography (SEC), and the detection of the exopolysaccharide molecules by refractive index (RI) can be used to estimate the polysaccharide amount in the corresponding elution peak [132]. To evaluate the polysaccharide composition, the glycosidic bonds must be previously hydrolyzed by enzymes or acid treatment. Thereafter, monosaccharide composition and quantity can be determined by high-performance liquid chromatography (HPLC) [124] or high-performance ion exchange chromatography with pulse amperometric detection (HPAEC-PAD) [133]. In order to have information not only on the composition but also on glycosyl linkages, a sample permethylated by the Ciukanu and Kerek method [134], hydrolyzed with trifluoroacetic acid, reduced with NaBH_4_ and acetylated can be analyzed by gas chromatography coupled with mass spectrometry (GC-MS) [133,135].

### 3.3. Nuclear Magnetic Resonance

NMR is a non-invasive analysis method frequently used to finely determine a polysaccharide’s chemical structure based on the radiofrequency radiation by nuclei in a magnetic field. 1D and 2D liquid NMR analyses have been used for the characterization of purified exopolysaccharides in biofilm [133,136]. However, this technique has low sensitivity and requires time-consuming purification of the analytes.

As an alternative to classical liquid NMR, solid-state NMR is a powerful tool that can be used to directly analyze intact cells or biofilm, providing quantitative determination [137,138]. Solid-state NMR approaches to characterize biofilms are essentially divided into bottom-up and top-down approaches: bottom-up approaches are employed when NMR spectra of matrix single components (usually not too many) are available. This approach serves to identify the extracellular matrix (ECM) composition of *E. coli*. Top-down approaches, used for *V. cholerae* ECM characterization, are used for more complex ECM and only require the intact ECM material, not relying on the samples of the isolated matrix component [138]. Both methods, however, are used for the analysis of static ECM, while, to understand the transformation of a planktonic cell to a biofilm, NMR-based metabolomic analyses have been applied to *S. aureus*, *E. coli* and *Pichia pastoris*: characterizing and comparing the metabolomic differences between planktonic cells and biofilms provide a means to identify active and relevant biological processes associated with biofilm formation [139,140]. A recent application was the combination of the SEM analysis (for biofilm architecture) and solid-state NMR (for biofilm composition) to characterize the biofilm of *A. fumigatus*. In particular, two types of one dimensional solid-state NMR experiments were performed: cross-polarization magic-angle spinning (CPMAS) and rotational-echo double resonance (REDOR) NMR. These experiments revealed that *A. fumigatus* ECM is rich in polysaccharides since anomeric carbons and ring sugar carbons account for about 43% of the total carbon mass. Through ^13^C[^15^N]REDOR, it was possible to identify and quantify carbons directly bonded to nitrogen, therefore allowing the characterization of the contribution of proteins to the extracellular matrix (ECM) [141].

### 3.4. Colorimetric Assays

Measurement of UV absorbance is typically used to detect protein and DNA, but this approach is not suitable for detecting polysaccharides and glycolipids due to the lack of specific chromophores, and prior chemical derivatization is therefore required. A wide range of colorimetric methods that specifically target polysaccharides has been developed, including Dreywood’s anthrone reagent [142], the phenol-sulfuric acid method [143], Monsigny resorcinol [144], Morgan–Elson method [145], Park–Johnson assay [146] and periodic acid Schiff assay (PAS) [147]. Each of these has peculiar properties regarding sample pretreatment, sensitivity, chemical interference and analysis time length.

Dreywood’s anthrone reagent assay is the classical colorimetric procedure to quantify polysaccharides. The major drawbacks of this assay are accuracy at concentrations higher than 10 μg/mL and high sensitivity to only traces of solvents used in the chromatographic separation [142,148].

Among the most popular procedures for the determination of exopolysaccharides and related compounds is the phenol-sulfuric acid method [143]. An interesting variation of this method allows the discernment between low molecular weight carbohydrates and polysaccharides by subtracting the reducing sugar fraction (RS) estimated by the dinitrosalicylic acid method [149] from the total sugar (TS) fraction measured by the phenol-sulfuric method [150]. A high-throughput phenol-sulfuric acid assay in a 96-well microplate format that allows longer linear range and excellent sensitivity was also optimized [151].

Monsigny resorcinol for neutral sugars [144], the Morgan–Elson method for free *N*-acetylglycosylamines [145] and the Park–Johnson assay for reducing sugars [146] have limited applicability to exopolysaccharides’ quantification due to their specificity.

The periodic acid Schiff (PAS) reagent has been traditionally used for histological staining of vegetal or mammalian tissues, and later found relevancy in polysaccharides’ quantification from the biofilm matrix. The PAS assay consists of a first oxidation step of the vicinal diols by the action of periodic acid, followed by the reaction of Schiff’s reagent with the newly generated aldehydes, leading to a purple-to-red product. The PAS method is three times more sensitive than the anthrone method [152], and in contrast to the above-mentioned assays, it does not face much interference from proteins, DNA or RNA [125]; it is able to capture neutral and charged carbohydrates; and it is not specific to any type of glycosidic linkage [153]. The 96-well microplate PAS assay has been optimized both on exopolysaccharides in solution or directly on miniaturized biofilms’ growth in microplate wells [125]. Additionally, the PAS assay needs a low sample volume, and, conversely to the phenol-sulfuric acid assay, it produces small amounts of non-hazardous waste, so it has the potential to be used as a high-throughput assay for biofilm quantification in clinical settings [153]. However, the limitation on the colorimetric response was shown for some carbohydrates probably due to a not-favorable configuration of reactive aldehydes after oxidation [147].

## 4. Therapies Targeting Exopolysaccharides

Given its complex and variable nature, the EPS matrix constitutes one of the main challenges for bacterial biofilm eradication [154]. The EPS matrix is considered as a barrier to overcome, due to the increasing resistance to antibiotics and their lack of efficiency on biofilms. Currently, clinical treatment on biofilm infections is still mainly based on antibiotics. The highly antibiotic resistant property of biofilms urgently requires potent antimicrobial agents and new antibiofilm approaches. Therapeutic strategies involving biofilm exopolysaccharides’ targets are mainly focused on destroying EPS through enzymes or compounds having activity against exopolysaccharides. These strategies generally help the diffusion of antibiotics into the biofilm. Research on immunotherapies to fight chronic infections and vaccines as a prevention strategy have also been reported. 

Therapies targeting exopolysaccharides in biofilms are summarized in Figure 2.

### 4.1. Enzymes

In a microbial biofilm, enzymes are naturally secreted by bacterial cells and retained within the matrix. Enzymes are crucial for the remodeling process of the biofilm, during which specific enzymes degrade EPS matrix components, bringing the active dispersion of the biofilm. Consequently, the dispersed cells can recolonize other sites of the host [155]. Enzymes are usually considered a biofilm virulence factor; however, their engineering may be used as a strategy for biofilm disassembly. Taking into consideration the importance of exopolysaccharides in the establishment and maintenance of biofilm architecture, glycoside hydrolases targeting exopolysaccharides in biofilms are widely studied.

Dispersin B (DspB) is a glycosyl hydrolase able to specifically disrupt PNAG, which is the most present exopolysaccharide of *S. epidermidis* biofilms [156]. On the other hand, PNAG is not one of the main components of *Burkholderia cenocepacia* and *Achromobacter xylosoxidans* biofilms, which can explain their higher resistance to the recombinant enzyme [24,157]. Moreover, recombinant dispersin B has been conjugated with a silver-binding peptide, which favors the in situ formation of silver nanoparticles (AgNPs) in the presence of silver ions [158]. In this combinatory strategy, the recombinant enzyme disaggregates the matrix and the AgNPs kill the dispersed cells. With respect to dispersin B alone, the enzyme–peptide conjugate displays at least a 2-fold higher activity against 48 hold *S. epidermidis* biofilms [159]. Finally, chitosan nanoparticles (NPs) have been designed for immobilization of dispersin B from *Aggregatibacter actinomycetemcomitans* HK1651. The in vitro antibiofilm efficacy of this formulation was assessed on *S. aureus*, *S. epidermidis* and *Aggregatibacter actinomycetemcomitans* 24 h-old biofilms [160]. For all tested strains, free and immobilized dispersin B exhibited an analogous disruptive effect on the biofilms, indicating that the immobilization of the enzyme into polymeric NPs did not worsen its activity.

Therapeutic strategies of *P. aeruginosa* biofilm infections include the Alginate lyase enzyme. Its ability to degrade Alginate, the key exopolysaccharide in the *P. aeruginosa* mucoid biofilm matrix, has been the supposed mechanism for biofilms’ disruption and has long been considered an encouraging approach to increase antibiotic efficacy [161,162,163]. Recently, it was demonstrated in an in vitro model that Alginate lyase dispersion of *P. aeruginosa* biofilms and enzyme synergy with antibiotic tobramycin are not correlated with catalytic activity [164]. These results provide new understanding into the potential mechanisms of Alginate lyase therapeutic activity, although additional work is needed to extend these data to more clinically relevant models. Moreover, Alginate lyase has been immobilized in polymeric nanoparticles (NPs) for matrix disruption purposes. In particular, ciprofloxacin-loaded NPs functionalized with Alginate lyase were designed for the treatment of biofilm-associated *P. aeruginosa* infection in cystic fibrosis [165]. The in vitro efficacy assay against 48 h-old biofilms displayed a full disruption of the EPS matrix and the killing of all bacteria after repeated treatment for 72 h. This effect was not seen with antibiotics alone or in combination with Alginate lyase and with non-functionalized NPs. Further, biofilm after treatment with the functionalized NPs was shown to be thin by a microscopy assessment. Additionally, in vitro and in vivo toxicity studies revealed that the developed nanosystem was biocompatible. Finally, silver nanocomposites were designed to deliver Alginate lyase and ceftazidime, which successfully disrupted biofilms and eliminated *P. aeruginosa* PAO1 from the mice lungs without evident side effects [166].

Glycoside hydrolases PelAh and PslGh [63], encoded in the *P. aeruginosa* Pel and Psl biosynthetic operons, respectively, have been shown to destroy biofilm integrity through degrading the main exopolysaccharides Pel and Psl within the biofilm matrix in vitro, respectively. Moreover, when *P. aeruginosa*-infected wounds were treated, the combination of PslGh with tobramycin enhanced bacterial clearance more than using tobramycin or PslGh alone. The combination can also promote activation of the innate immunity, resulting in greater complement deposition, neutrophil phagocytosis and neutrophil reactive oxygen species production [167]. Recently, it has been demonstrated that PelAh, once immobilized on bacterial cellulose (BC), can be used in BC wound dressings for safe and specific protection against biofilm formation by *P. aeruginosa* [168].

Table 3 summarizes some of the glycoside hydrolases with biofilm-disrupting ability.

### 4.2. Inhibitors of Exopolysaccharides’ Production

It has been reported that some compounds are able to inhibit the production of the exopolysaccharide components of the biofilm matrix. For instance, the antifungal caspofungin acetate was demonstrated to be active against methycillin-resistant *S. aureus* (MRSA) biofilms. This lipopeptide is effective in bacterial biofilms as a PNAG synthesis inhibitor, by impeding the enzymatic activity of IcaA operon, which shares homology with β-1-3-glucan synthase (caspofungin’s pharmacological target in fungi). In combination with fluoroquinolones antibiotics, caspofungin acetate displayed potential synergistic effects, both in vitro and in vivo systems for catheter-based infections [182].

Ambroxol is a commonly used mucolytic agent with antioxidant and anti-inflammatory effects in patients with pulmonary infections [183,184]. A combination of ambroxol with ciprofloxacin was studied in a rat model of *P. aeruginosa* acute lung infection, demonstrating that ambroxol has a synergistic effect with ciprofloxacin on mucoid biofilms [185]. This effect could be attributable to the direct activity of ambroxol against Alginate, which is the principal constituent of these biofilms, by increasing the expression of the *mucA* gene; decreasing the expression of *algD*, *algU* and *algR* genes; and reducing activity of diphospho-d-mannose dehydrogenase (GDP-mannose dehydrogenase; GMD) [186]. These genes and enzymes are critical to the formation of Alginate.

Alginate inhibitors, such as thiol-benzo-triazolo-quinazolinones, have been demonstrated to hamper the binding of bis(3′-5′)-cyclic dimeric guanosine monophosphate (c-di-GMP) to the receptor protein Alg44, resulting in a decreased Alginate biosynthesis in *P. aeruginosa*, which could potentially limit mucoid conversion and reduce a complication in CF patients [187]. The more potent thiol-benzo-triazolo-quinazolinone has the ability to reduce *P. aeruginosa* Alginate secretion up to 30%. These compounds could be used as leads in the development of novel inhibitors of Alginate.

High-throughput gene expression screening approaches have been used to search for new molecules impacting biofilm formation, in particular to identify compounds that repress expression of the Pel genes in *P. aeruginosa*. The Pel repressors demonstrated in vitro antibiofilm and antivirulence activities. Some of the repressors’ compounds were structurally similar and shared a benzothiazole scaffold. These compounds did not prevent the bacterial growth but reduced the expression of the Pel gene, thus affecting the biofilm growth [188]. These small molecules represent a new anti-infective approach for the possible treatment of chronic *P. aeruginosa* infections.

### 4.3. Immunotherapies

Novel antibody-based therapies may represent a strategy to target biofilm-specific components for host-cell mediated bacterial clearance. For *P. aeruginosa*, human monoclonal antibodies (mAbs) targeting three distinct Psl exopolysaccharide epitopes [189] (class I, II and III mAbs) are able to inhibit early biofilm events and promote bacterial clearance from preformed biofilms in the presence of host-effector cells [190,191]. Enhanced anti-*P. aeruginosa* activity was achieved by multimechanistic bivalent, bispecific antibody configuration targeting a Psl exopolysaccharide and PcrV protein [192]. This bispecific antibody, called MEDI3902 (Astrazeneca), is currently in phase 2b development for prevention of nosocomial pneumonia in patients undergoing mechanical ventilation. The whole-genome sequencing on 913 isolates from diverse patient populations and geographical locations allowed the statement that Psl and PcrV are highly prevalent in global clinical isolates, suggesting MEDI3902 can mediate broad coverage against *P. aeruginosa* [193].

Given its ubiquitous expression and its role as a key virulence factor, PNAG has been for a long time considered a potential universal polysaccharide antigen [29]. However, natural antibodies to PNAG are not functional in mediating in vitro microbial killing or in vivo protection. Conversely, deacetylated PNAG (dPNAG) poly- and oligosaccharides conjugated to carrier proteins were shown to elicit immunity across several microorganisms [194]. A protective effect was demonstrated in preclinical models for the human monoclonal antibody IgG F598, able to bind both native PNAG and dPNAG, which reached human Ph II testing. In particular, by hybridoma technology, three mAbs (both as IgG1 and as IgG2) were produced from B cells of an individual post-*S. aureus* infection. The three mAbs were all equally able to bind to native PNAG, but mAb F598 bound best to non-acetylated or backbone epitopes on PNAG and was preferred to F628 and F630 that bound optimally to PNAG and minimally to dPNAG. For this reason, F598 was selected as a clinical candidate [195].

### 4.4. Vaccines

Vaccines can be used as a prevention strategy against clinical infection. However, there are currently no vaccines targeting exopolysaccharides of the biofilm matrix approved for human use.

All clinical trials of *P. aeruginosa* vaccine candidates have failed to date [196]. Individuals with cystic fibrosis (CF) who are not colonized with mucoid *P. aeruginosa* appear to produce opsonizing and phagocytic antibodies against Alginate [197]. While beneficial, these antibodies are usually not sufficient to eradicate the bacteria upon exposure, and prevent the establishment of future chronic infection [198]. For this reason, an enhancement of the immune response against Alginate makes it an appealing candidate for therapeutic and prophylactic vaccination. Moreover, as Alginate has very little structural variation, it is considered as an ideal target for developing new vaccines. In a phase I study assessing a range of Alginate extracts from a mucoid strain of *P. aeruginosa*, extracts of a larger molecular weight induced opsonizing antibodies for up to two years post-vaccination; however, the overall immunogenicity across the study was poor [199]. This can reflect an essential property of the Alginate, wherein any protective epitopes are poorly immunogenic, allowing chronic infection to persist in the otherwise immunocompetent CF host. Although antibodies to Alginate are found in the sera of chronically infected CF patients, these antibodies lack the mediation of the opsonic killing of *P. aeruginosa* in vitro [200]. Several Alginate vaccines for *P. aeruginosa* have also been reported at a preclinical stage. With the aim to improve its immunogenicity, Alginate has been conjugated to carrier proteins, such as keyhole limpet hemocyanin (KLH) [201], *P. aeruginosa* toxin A [202], tetanus toxoid (TT) [203], type b-flagellin (FLB) [204], outer membrane vesicle (OMV) of *Neisseria meningitidis* serogroup B [205] and recently to PLGA nanoparticles [206]. All of these glycoconjugates tested in mice animal models gave good immunological results.

Moreover, a TT-conjugate vaccine based on the synthetic de-*N*-acetylated PNAG pentasaccharide (5GlcNH_2_-TT) was tested on more than ten infection models and it progressed to clinical trials, showing good promise for a synthetic glycoconjugate vaccine against PNAG-producing pathogens [194,207].

## 5. Future Perspectives

Throughout evolution, bacterial species have developed the adaptability to form biofilms, to tolerate and survive in extreme conditions such as mechanical stress and antibiotic treatments. Biofilm-associated infections are currently a main health concern. Many chronic infections are related to the formation of bacterial biofilms, and conventional antibiotic treatment is often not adequate to eliminate biofilm-related infections [208]. Several strategies have been explored to prevent biofilm formation; however, a proper and efficient approach has not been found yet because microorganisms evolve dynamically when exposed for a long time to antibiotics, achieving tolerance to them. An in-depth understanding of biofilm formation, antibiotic resistance mechanisms and host–biofilm interactions is pivotal to novel discoveries in the field.

One of the principal components of the biofilm matrix are exopolysaccharides which play a main role in keeping the biofilm structure; therefore, targeting exopolysaccharides could be helpful for the complete eradication of biofilms. As an example, the use of disrupting enzymes such as polysaccharide lyases together with conventional antibiotics has been shown to be helpful in disaggregating biofilms and making the bacteria susceptible to antibiotic treatment [162,164]. Usually, biofilm composition consists of several bacterial species’ populations; this makes the antibiotic treatment inefficient, which would otherwise work on single-species biofilms. In polymicrobial biofilms, the exopolysaccharides producing bacteria have the ability to confer resistance to non-exopolysaccharides-producing microorganisms, and proximity favors the transfer of antibiotic-resistant genes. Therefore, studies regarding cell-to-cell interactions within polymicrobial biofilms and also with different antimicrobial agents are needed to develop an effective therapy that would work on the entire biofilm. For this purpose, combinatory therapies targeting different components of the matrix have increasingly gained attention.

In conclusion, the ideal approach for biofilm eradication would combine antibacterial agents with strategies for EPS disassembly able to increase cell susceptibility to antibiotics [154]. Therefore, this innovative therapy would decrease bacterial resistance and prevent biofilm recurrence.

## Figures and Tables

**Figure 1 ijms-24-04030-f001:**
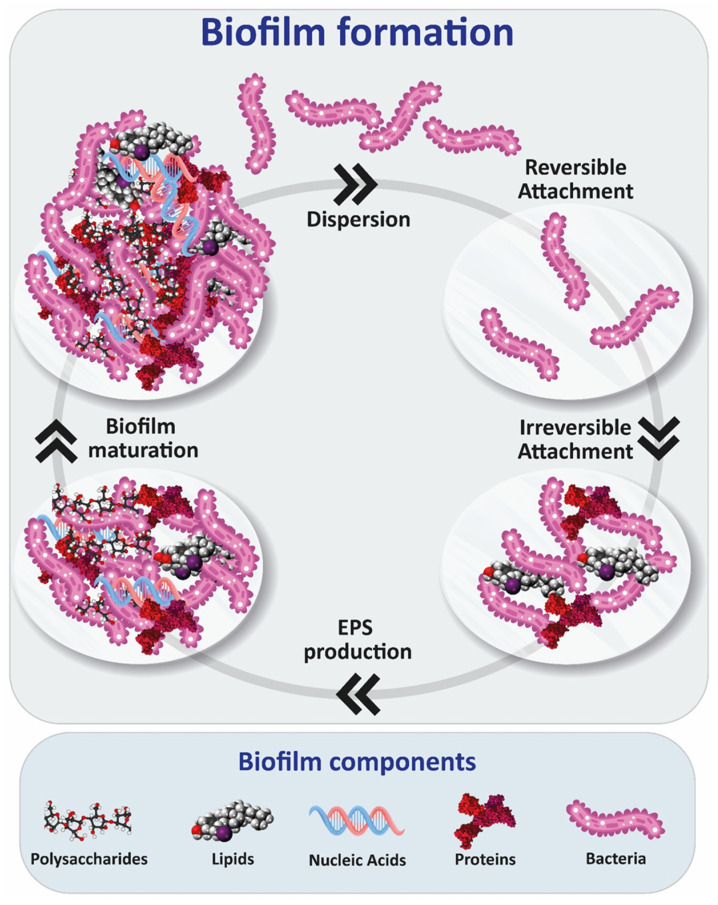
Stages of biofilm formation: reversible attachment, irreversible attachment, EPS production, maturation and dispersion/detachment phase.

**Figure 2 ijms-24-04030-f002:**
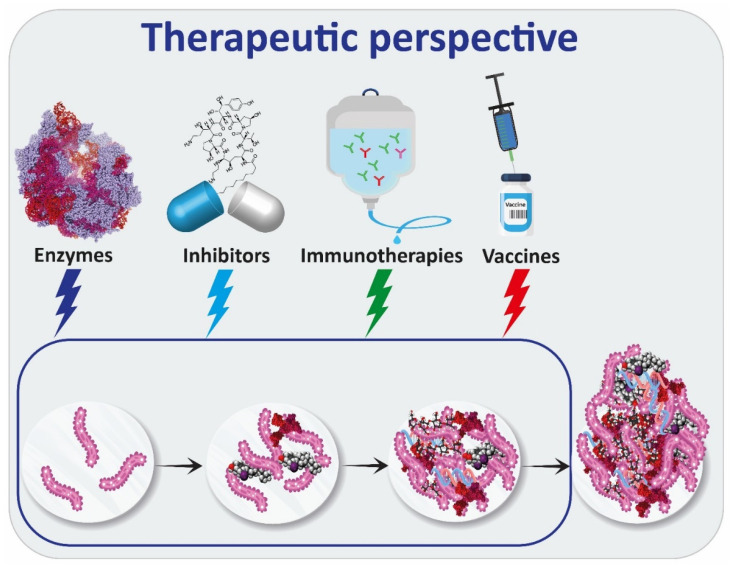
Summary of therapies targeting exopolysaccharides in biofilms.

**Table 1 ijms-24-04030-t001:** Polysaccharides in biofilm and their function.

Polysaccharide	Pathogen	Function	References
PNAG	*S. aureus*, *S. epidermidis*, *E. coli*, *Y. pestis*, *A. pleuropneumoniae*, *A. baumannii*	Attachment of bacterial cells to a surface, biofilm proliferation and maturation, final detachment	[20,21,22,23,29]
Alginate	*P. aeruginosa*	Biofilm maturation, formation of thick and highly structured biofilms with differentiated microcolonies, protection against antibiotics, resistance to oxidative stress	[39,40,41]
Psl	*P. aeruginosa*	Adhesion to surface and cell-to-cell interactions, role in aggregation of microcolonies, maintenance and maturation of biofilm through promotion of c-di-GMP production, shielding from antimicrobials and neutrophils	[39,40,41]
Pel	*P. aeruginosa*	Surface attachment, structural scaffold, confers tolerance to aminoglycoside antibiotics and to colistin	[39,40,41]
Galactosaminogalactans	*A. fumigatus*	Adherence to surface, anti-inflammatory properties	[91,92,93,94,95]
Galactomannans	*A. fumigatus*	Silence inflammation, promote the initial establishment of fungal cells in the lung	[91,92,93,94,95]
Glucans	*A. fumigatus*, *S. mutans, Candida*	Resistance mechanism to antifungal drugs, adhesion, agglutination mediation	[91,92,93,94,95,105,106,122]
Fructans	*S. mutans*	Adhesion	[105,106]
Vibrio polysaccharide (VPS)	*V. cholerae*	Determines biofilm architecture, cell–cell adhesion, stability	[102]
Colanic acid	*E. coli*	Attachment to abiotic surface	[107,109,110]
Cellulose	*E. coli, Salmonella*	Adherence to abiotic surface, cell-to-cell interaction, biofilm architecture	[107,112,113]
Wall teichoic acids (WTAs)	*S. aureus, S. epidermidis*	Adhesion of bacteria to surfaces and other bacterial cells	[115,118]
Xanthans	*X. campestris* *pathovar campestris*	Suppression of plant defenses, attachment initiation	[120,121]

**Table 2 ijms-24-04030-t002:** List of lectins for the characterization of monosaccharides and disaccharides in biofilm polysaccharides.

Source	Full Name	Abbreviation	Specificity
Lectin from mushrooms	Aleuria Aurantia Lectin	AAL	Fucose; Arabinose [a]
Lectin from seeds	Amaranthus Caudatus Lectin	ACA/ACL	Galactose-β(1-3)-N-acetylgalactosamine [b]
Lectin from ground elder	Aegopodium Podagraria Agglutinin	APA/APP	N-acetylgalactosamine [c]
Lectin from chickpeas	Cicer Arietinum Lectin	CAA/CAL	N-glycans; N-acetylgalactosamine [b]
Lectin from jack-bean	Concavalin A	ConA	α-Mannose; α-Glucose [a]
Lectin from coral tree seeds	Erythrina Cristagalli Lectin	ECA/ECL	β-Galactose; N-acetylgalactosamine; Lactose [a]
Lectin from snowdrop bulbs	Galanthus Nivalis Lectin	GNA/GNL	α-(1,3)-Mannose [a]
Lectin from seed	Griffonia Simplicifolia Lectin I	GSL-I/BSL-i	Galactose; N-acetylgalactosamine [a]
Lectin from roman snail	Helix Pomatia Lectin	HPA/HPL	α-N-acetylgalactosamine [d]
Lectin from lima bean	Phaseolus Limensis Agglutinin	LBA	N-acetylgalactosamine-α(1,3)-Galactose [e]
Lectin from lentil	Lens Culinaris Agglutinin	LCA	α-Mannose; α-Glucose [a]
Lectin from tomatoes	Lycopersicon Esculentum Lectin	LEL/LEA	N-acetylglucosamine [a]
Lectin from asparagus pea	Lotus Tetragonolobus Lectin	LTL	Fucose; Arabinose [a]
Lectin from horseshoe crab	Limulus Polyphemus Lectin	LPA/LPL	Sialic acid; N-acetyl-d-hexosamines; D-glucuronic acid [f]
Lectin from winged pea	Lotus Tetragonolobus Lectin	LTL	Fucose; Arabinose [a]
Lectin from daffodil bulbs	Narcissus Pseudonarcissus Lectin	NPA/NPL	α-Mannose [a]
Lectin from bacteria	Pseudomonas Aeruginosa Galactophilic Lectin	PA-I	Galactose [g]
Lectin from beans	Phytohaemagglutinin	PHA	Different oligosaccharides [h]
Lectin from peanuts	Peanut Agglutinin	PNA	Galactose; N-acetylgalactosamine [a]
Lectin from peas	Pisum Sativum Agglutinin	PSA	α-Mannose; α-Glucose [a]
Lectin from ricinus	Ricinus Communis Agglutinin	RCA	Galactose; Lactose [a]
Lectin from soybeans	Soybean Agglutinin	SBA	Galactose; N-acetylgalactosamine [a]
Lectin from potatoes	Solanum Tuberosum Lectin	STA/STL	N-acetylgalactosamine [a]
Lectin from gorse	Ulex Europaeus Agglutinin I	UEA I	α-Fucose [a]
Lectin from fava beans	Vicia Faba Agglutinin	VFA	Mannose; Glucose [h]
Lectin from hairy vetch seeds	Vicia Villosa Lectin	VVA /VVL	N-acetylgalactosamine [a]
Lectin from Japanese Wisteria seeds	Wisteria Floribunda Lectin	WFA/WFL	N-acetylgalactosamine [a]
Lectin from wheat germs	Wheat Germ Agglutinin	WGA	N-acetylglucosamine; Sialic acid [a]

Specificity data retrieved for vendor website: [a] Vector Laboratories; [b] CliniSciences; [c] CD BioGlyco; [d] Thermo Fisher; [e] GlycoMatrix; [f] EY Laboratories; [g] Fisher Scientific; [h] Merck.

**Table 3 ijms-24-04030-t003:** Glycoside hydrolases that disperse biofilms. Adapted with permission from [169].

Enzyme	Bacteria Where It Acts	Summary	References
Dispersin B	*S. aureus*, *A. actinomycetemcomitans, S. epidermidis*, *A. baumannii*, *K. pneumoniae*, *E. coli, Burkholderia* spp., *Actinobacillus Pleuropneumoniae*, *Yersinia Pestis, Pseudomonas fluorescens*	Produced by *A. actinomycetemcomitans*, it degrades the polysaccharide PNAG through hydrolyzing β(1,6) glycosidic linkages.	[24,170]
Alginate lyase	*P. aeruginosa*	It degrades the exopolysaccharide Alginate, causing bacterial cell dispersal and increasing antibiotics’ efficacy and phagocytosis.	[162,164]
PelAh, PslGh	*P. aeruginosa*	They disperse mature biofilms by hydrolyzing Pel or Psl exopolysaccharides, respectively.	[63]
Cellulase	*S. aureus* and *P. aeruginosa*	It is produced by multiple microbes and it hydrolyzes the β(1,4) glycosidic linkage.	[171]
α- mannosidase	*P. aeruginosa*	It is believed that this acid hydrolase is involved in the turnover of N-linked glycoproteins. However, it has a cytotoxic effect on A-431 human epidermoid carcinoma cell lines.	[172,173]
β- mannosidase	*P. aeruginosa*	It hydrolyzes the terminal mannose residues, which are β(1,4) linked to oligosaccharides or glycopeptides. However, it has a cytotoxic effect on A-431 human epidermoid carcinoma cell lines.	[172,174]
α -amylase	*V. cholerae*, *S. aureus* and *P. aeruginosa*	It can derive from multiple sources and hydrolyzes α(1,4) glycosidic linkages, mediating the dispersal of mature biofilms of multiple bacterial strains.	[171,175,176]
Hyaluronidase	*S. aureus* and *S. intermedius*	It cleaves hyaluronic acid (HA), a component which has been found to be incorporated into the biofilms formed by multiple pathogens, causing biofilms’ dispersal.	[177,178]
PgaB	*B. pertussis*, *Staphylococcus carnosus, S. epidermidis* and *E. coli*	It disrupts PNAG-dependent biofilms through hydrolyzing PNAG, a major biofilm component of many pathogenic bacteria.	[179]
Ega3	*Aspergillus fumigatus* and *P. aeruginosa*	This endo-acting α1,4-galactosaminidase disrupts biofilms formed by GAG-dependent *Aspergillus fumigatus* and Pel polysaccharide-dependent *P. aeruginosa*.	[180]
Sph3	*A. fumigatus*	This retaining endo-α-1,4-N-acetylgalactosaminidase hydrolyzes galactosaminogalactan (GAG), a cationic polymer of α-1,4-linked galactose and partially deacetylated N-acetylgalactosamine (GalNAc), causing biofilm disruption.	[181]

## Data Availability

No new data were created or analyzed in this study. Data sharing is not applicable to this article.

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
