# Peer review of "Polysaccharides’ Structures and Functions in Biofilm Architecture of Antimicrobial-Resistant (AMR) Pathogens"

_ijms, 2023, doi:10.3390/ijms24044030_

Round 1
Reviewer 1 Report
I have gone through the manuscript ijms-2207179 entitled “Polysaccharides structures and functions in biofilm architecture of AMR pathogens”. The authors describe the structure and function of extracellular polysaccharides in great detail. Whole manuscript is very well organized with every subheading being extensively explained in details. There are only some small technical details that should be addressed. I recommend this manuscript for publication after minor revision.
1. Throughout the whole manuscript please put all bacterial and fungal strains names into italic form.
2. In the introduction section, line 34, please give the full names of every bacterial species. It is a common when a bacterium is first mentioned in text that the whole name of species is used (for example Pseudomonas aeruginosa) and after, the P. aeruginosa can be used in the rest of the text.
3. In the introduction section, line 67, please give full name of PNAG in brackets.
4. Lines 92-97, this sentence is a bit longer and confusing, please redefine it in order to be more understanding.
5. The name of the locus/operon should be in italic form as well (line 98, pga locus and line 113, ica operon, line 1085).
6. Lines 130 and 132 change ica to ica
7. Throughout the whole manuscript please put in vitro and in vivo into italic form.
8. Lines 200-202, it would be more understanding if this sentence is split into two.
9. Line 279, please change genes codifying into genes coding.
10. Line 363, add Salmonella spp. or Salmonella species.
11. Line 404, change S. Aureus to S. aureus
12. Throughout the whole manuscript please put references before a period or comma, not after.
Author Response
Thanks for the review and the nice comment. Below the response to each comment:
1) Bacterial and fungal strains name have been written in italic
2) Bacteria species name have been changed accordingly throughout the whole manuscript
3) Line 67: full name of PNAG has been provided
4) The sentence has been rephrased as follows: "PNAG biosynthesis pathway has indeed been characterized in different species: in S. aureus and S. epidermidis PNAG is synthesized by proteins encoded by the icaADBC genes of the intercellular adhesin locus (ica), while E.coli has a homologous genetic locus, named pgaABCD [22,27]."
5) The name of locus operons have been put in italic
6) The change was made
7) In vitro and in vivo have been put in italic
8) The sentence has been split in two
9) Correction was made
10) spp. was added
11) S. aureus was corrected
12) Reference position was changed throughout the manuscript
Reviewer 2 Report
The manuscript entitled “Polysaccharides structures and functions in biofilm architec-2 ture of AMR pathogens”, is of interest for the readers. Below please find my suggestions and comments regarding with your paper:
- The title : authors should replace the abreviation 'AMR' with full worlds because it was not defined in the text.
- Illustration : only one figure was presented in this review, which is not enough. More illustrations are needed.
- Why authors treated the potential new antimicrobial therapies for biofilm formation inhibition? I think it can be anothor review apart.
Author Response
Many thanks for the comment and the suggestion.
- Title: The word AMR in the title has been replaced with Antimicrobial resistant
- Illustrations: there are already two figures in the review, one in the introduction and one in the therapeutic strategies. As there are two tables, we preferred not to include additional figures.
- New antimicrobial therapies were included in the review because we aimed not only to provide a characterization of the polysaccharides that are forming the biofilm structure, but also to suggest potential strategies to target them as new ways to tackle antimicrobial resistance. Presenting only the characterization of the biofilm without linking it to the development of new antimicrobial strategies was not coherent with the topic of the special issue: "Antimicrobial Resistance, Molecular Mechanisms and Fight Strategies 2.0".
Reviewer 3 Report
Journal: IJMS (ISSN 1422-0067)
Manuscript ID: ijms-2207179
Type: Review
Title: Polysaccharides structures and functions in biofilm architecture of AMR pathogens
Authors: Evita Balducci * , Linda Del Bino * , Francesco Papi , Daniela Eloisa Capialbi
Section: Molecular Microbiology
Special Issue: Antimicrobial Resistance, Molecular Mechanisms and Fight Strategies 2.0
Comments
With AMR pathogens as targets, all efforts are made to either kill them or suppress their pathogenicity.
The article describes the physiological characteristics of the biofilm. Matrix is an important target, but equally important are the other pathogenicity factors which are genetic in nature. The manuscript is skewed and may not provide information, which may be sufficient to treat infectious diseases. Antipathogens should also have been considered towards the end as a complementary mechanism to fight pathogens.
Minor issues:
Lines 33-34:
Both Gram positive and Gram negative bacteria can form biofilms, but the main concerns are raised regarding E. faecalis, S. aureus, S. viridans, E. coli, K. Pneumoniae, P. aeruginosa. [2,3]
Q: Most accepted statement regarding pathogens of concern is:
The ESKAPE pathogens (Enterococcus faecium, Staphylococcus aureus, Klebsiella pneumoniae, Acinetobacter baumannii, Pseudomonas aeruginosa, and Enterobacter species) are the leading cause of nosocomial infections throughout the world.( Biomed Res Int. 2016; 2016: 2475067. doi: 10.1155/2016/2475067 and many other References support this statement)
Lines 36-52: Formation of a mature biofilm ……………
Q: There seems to be no need to present the conventional and most recent version of the formation of a mature biofilm. The authors can describe only one.
Line 93: “if” seems to be redundant
Line 321: “born” seems to be incorrect.
Line 316: 2.4 Other polysaccharides
The material under this heading needs to be segregated under two subheadings: bacteria and fungi
Line 84 onwards:
2. Biofilm polysaccharides:
Q: Too much focus is on characterizing polysaccharides. The text matter can be summarized into two brief paragraphs.
Line 420 onwards:
3. Characterization of exopolysaccharides:
Q: Too much focus is on characterizing exopolysaccharides. The text matter can be summarized into two brief paragraphs.
Author Response
Many thanks for the comment and the suggestions to improve our manuscript.
We are aware that other genetic factors are as important as the biofilm matrix for pathogenicity, and that antipathogens are still the main therapeutic approach and we don't want to disregard them. However, we have realized that among the different anti AMR targets and strategies, extracellular matrix has been poorly taken into consideration, especially the exopolysaccharide components. We ourselves felt the need to collect literature evidences that have been published on bacterial exopolysaccharides in biofilm in the last years since this topic has not been described in-depth so far in a single article.
1) Sentence 33-34 was rephrased as suggested including the reference to the ESKAPE pathogens
2) Lines 36-52: the definition and description of the new biofilm model was removed and only the reference is left
3) Line 93: the whole sentence was rephrased and "if" removed
4) Line 321: "born" was corrected to "borne"
5) Line 316: in the last part of the paragraph "Other polysaccharides" the examples related to fungi have been segregated
6) Line 84 onwards: bacterial polysaccharides are a well-known vaccine target, however their role in biofilm have not been extensively studied. In other reviews they are often described briefly and in a poorly detailed way. Therefore, we believe it is interesting to treat extensively this part in the paragraph. We aimed to collect the information reported in different research papers and made them easier to access by whom is working in the field or is interested in this topic since we believe this can represent a significative contribution and a push forward the research on new antimicrobial strategies.
7) Line 420 onwards: since in the current literature reviews on the exopolysaccharide characterization and quantification in biofilm do not exist, we believe it is interesting to include this part. Not many details are provided on each single tool but we aimed to list all the different techniques that can be applied (also in a complementary way) to exopolysaccharide characterization. Our goal was to merge in a single paragraph the fragmented information reported in several research papers and make them more accessible to all the research community.
Round 2
Reviewer 3 Report
Accept